# Naturalistic Elements in Percival Everett's *Wounded*

Leah Abuan Milne 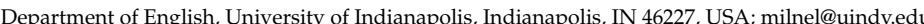

Department of English, University of Indianapolis, Indianapolis, IN 46227, USA; milnel@uindy.edu

**Abstract:** This article examines Percival Everett's 2005 novel *Wounded* through three applications of the word "naturalistic" in order to show how the work complicates divisions between nature and man and man and animal. First, the article shows how its protagonist, John Hunt, contends with his relationship to nature as both a source of respite and as part of his livelihood as a horse trainer. Next, "naturalistic" elements of Wounded reveal how the mythic West of classic Westerns has influenced perspectives on the "real" West. Finally, the article assesses *Wounded* as a work of American literary naturalism, particularly in terms of the questions it inspires about free will and determinism. Together, these applications of the word "naturalistic" expose how nature acts as John's barometer for human morality and individualism, forcing us to question whether man is truly superior over nature.

**Keywords:** Percival Everett; wounded; naturalism; African American literature; ecocriticism

> "The working title is *Wounded*. It's a really naturalistic novel. My interest is in the form of a realistic novel. You have to love the form you're working in, but I'm seeing what I can do."
>
> Percival Everett

## 1. Introduction

Percival Everett's seemingly offhand disclosure—in this epigraph and an interview with fellow fiction writer Rone Shavers—caused me to do a double take. Everett's description of his 14th full-length novel, *Wounded* (Everett 2004b), as "naturalistic" brought three possible meanings to mind. First, without context, a superficial assumption might be that Everett is simply describing the natural themes and setting of *Wounded* as "naturalistic". After all, *Wounded* is a Western whose stark environment places it alongside Everett's other novels and stories taking place in American deserts, including his parody of the Western genre in *God's Country* (Everett 2003a), and his more realistic portrayal of nature in works like "Alluvial Deposits" (Everett 2004a) and *Telephone* (Everett 2020). This straightforward application of "naturalistic" is clear when the protagonist and narrator of *Wounded*, John Hunt, a Berkeley-educated, art-loving rancher is asked why he chose to be a horse trainer in the fictional Highland, Wyoming, among mostly white neighbors. His answer is to simply gesture towards the dramatic natural landscape of mountain ranges and prairie grasslands. Standing at the top of a ridge overlooking the Red Desert, John explains to his friend's son and eventual ranch hand, David, "This is why I live here, . . . Every time I come up here and look at that [desert] I know my place in the world. It's okay to love something bigger than yourself without fearing it. Anything worth loving is bigger than we are anyway" (Everett 2005, p. 134). As seen in novels such as *Watershed* (Everett 2003c), *American Desert* (Everett 2004c), and *Assumption* (Everett 2011), the natural West of Everett's novels is sometimes brutal and harsh but mostly majestic and awe-inspiring.

A second, more subtle application of the word "naturalistic" can be found in the "-ic" suffix, which implies not a direct representation, but something with a close correlation to the word that precedes it—in this case, something having the character of nature or the natural. Given the sometimes performative nature of this representation, this second

definition of naturalism appropriately evolves from the theater; for instance, Sarah Balkin defines naturalism in this context as a "heightened form of realism" (Balkin 2019, p. 17). As a person who often observes his world as though outside of it, John's experiences indeed resemble a simulacrum of the natural, with a cast of characters delivering lines as though on a stage. For instance, his eventual love interest, a neighboring rancher named Morgan, teases John on his laconic sensibility and his fierce aversion to discussing anything sexual, romantic, or overly emotional, a stance likely influenced by the traumatic death of John's wife, Susie, six years earlier. Throwing the words "woman" and "sex" at him to gauge his reaction, an amused Morgan turns to John's uncle and roommate Gus, stating, "He's tenser than a Republican with a thought of his own." Rather than quietly admiring Morgan's wit, John "looked at Morgan, frowning a smile. 'Where'd that come from?'" Morgan, taking no offense, simply tells him that she'd "been waiting to use" the line for a while (Everett 2005, p. 31). To John's relief, Morgan's abstract rehearsal of the line and its execution take over the discussion, diverting it momentarily away from a subject John would rather avoid. This kind of meta-dialogue, a hallmark of much of Everett's autodiegetic narration in his other novels, here becomes part of the everyday interactions of many of the characters. Despite Everett's comment in the epigraph that he seeks realism in *Wounded*, this element of the novel shows how this work is adjacent to his other more surrealist and metafictional texts.

However, dropped in the middle of an interview on genre and aesthetics, Everett's description of *Wounded* encourages consideration of "naturalistic" in a third sense, having to do with an association with the fin de siècle literary movement of the same name. We have Émile Zola to thank for naturalism as a literary term, when Zola rather insistently defined it in his essay, "The Experimental Novel," as a movement within realism, so much so that its approach is scientific. "I have almost worn myself out repeating that naturalism is not in the words; that its strength lies in the fact that it is a scientific formula," Zola proclaims. "How many times shall I be obliged to say again that it is simply the study of men and things by observation and analysis, entirely free from any preconceived idea of the absolute?" (Zola 1893, p. 93). Zola may have been annoyed that his American contemporaries—Frank Norris, Stephen Crane, Theodore Dreiser, Jack London, Kate Chopin, and even later authors such as Richard Wright—imported an American aesthetic of naturalism that skirted the lines between realism and romanticism, shifting Zola's scientific method to characters whose lives are often ruled by fate and who must either adapt and compromise their ideals or risk annihilation. Frederick Jackson Turner, whose frontier thesis influenced much of our thinking of both the Western and of American literary naturalism, noted that, particularly in the less regulated areas of the West, society became diffuse, dissolved into its individual atoms, at the same time that tradition, precedent—in a word, the past—lost its power by this migration into the new world beyond the mountains. ... the simplicity of life... together with the vast extent of unoccupied land and unexploited resources, made it easy for [an] upcountry democrat to conceive of equality and competitive individualism as consistent elements of democracy. (Turner 1935, p. 20) With new (to the settlers) territory came both optimism and rootless instability, which infiltrated the literature of the time.

In this way, many critics feel that the conventions of American literary naturalism—which at its height roughly spanned the mid- to late-1800s through the 1910s—diverged from the philosophical center that ordered Zola's French version, instead organically responding to swiftly changing historical conditions with vacillating beliefs in pessimistic determinism and Emersonian proclamations of free will and individualism leading either to destruction or to moral and social progress. As Eric Carl Link notes, American literary naturalists "share a common desire to try to make sense of human nature in the wake of the scientific and philosophical revelations and revolutions of the nineteenth century" (Link 2012, p. 89). This desire thus worked under the guise of realist- and scientifically-inspired representations: In naturalist texts, characters who are often but not always disorderly and lower-class struggle fruitlessly against powerful hereditary, instinctual, and social inclinations. Works by authors like Norris and London represent the more optimistic end of this naturalistic aesthetic, with protagonists sometimes gaining at least some power

over humanity and nature as they make their way West, encountering Wild West frontiers and/or wild urban cities, the latter being a hopeful nod to human progress. However, even their works challenged the dominant narrative of the US as the land of freedom and opportunity.

Everett's novel, I argue, combines these three different approaches to naturalistic aesthetics, a move possibly inspired by the seemingly unforgiving environment of the desert itself. Especially when it comes to identity, Everett's work often dissolves categorical binaries in favor of multi-valued layers of humanity. *Wounded* does not collapse all binaries, but the ones it leaves intact—most notably, the line separating human animals and non-human animals—reveals how the novel complicates shared otherness or, when thinking of John's connection to nature in particular, shared animal-ness across the human/non-human divide. For John, nature acts as not just a respite and a retreat, but also as a barometer for human morality, and a potential answer to the question of whether humans have free will or not. This article will thus examine *Wounded*'s naturalistic elements in all three of the above senses of the term, showing how the three meanings interact to paint a conflicted picture of the desert frontier and its human residents.

## 2. Nature and Naturalism

The first and most obvious application of "naturalistic" is as a descriptor pertaining directly to nature. This is particularly evident in the American version of literary naturalism, whose authors, Link notes, "engage, at the thematic level, post-Darwinian reconsiderations of the relationship between humans and nature" (Link 2012, p. 73). *Wounded* immediately complicates the dualism of nature versus humanity and human animal versus non-human animal, showing how human intervention in nature both complicates the nature/human binary while ultimately exposing the damage that people can enact when they do not listen and adapt to nature's demands. Such distinctions highlight what Kevin Trumpeter calls "the defining preoccupation of literary naturalism": namely, its "suspicion toward inherited notions of human-centered agency" (Trumpeter 2015, p. 225). "Nature" is often perceived as that which is untouched by humanity, thus associated with purity and pristine environments. As Jon Mooallem observes, this binary ceased to exist as soon as human beings appeared; in other words, the nature/human binary is a man-made fiction:

> When we look at nature, maybe most of what we see is lines that we've superimposed there ourselves: taxonomic lines, legal lines, baselines of how we believe the world is supposed to look. These lines have only as much authority as we give them. It's our emotions that fasten them in place: how deeply we *believe* in their truths, and how guilty or queasy we feel when we come close to crossing them, even when crossing them may be the rational thing to do. Maybe the most imposing line is the one we imagine between ourselves and nature—the belief that there is such a thing as pristine nature, and that it is sacrosanct, and that any changes we trigger in it can only be disfiguring. (Mooallem 2013, pp. 191–92)

Mooallem's work suggests that, while the lines between humans and nature are arbitrary, our imposition of purity on nature without human interference leads us to misunderstand how we are part of and therefore responsible to our fellow natural things. *Wounded* features many of the lines mentioned above, both literally in terms of fences and roadways as well as figuratively and emotionally, as John reflects on where and how he intervenes in nature. His ranch also represents a legal line in terms of his personal property and the adjacent tribal and Bureau of Land Management (BLM) lands, some of which he owned and sold, and others that he leased after quitting cattle ranching. How John and others categorize aspects of nature also lends them to assign them value and notions of authenticity and purity, even as all creatures continue to evolve and adapt to natural and man-made changes in the landscape.

The arbitrariness in which humans have written themselves out of, into, or even above nature is evident in *Wounded*'s first lines. Narrated by John, the novel begins, "By definition a cave must have an opening large enough to allow a human to enter. The cavity can be

wind- or water-eroded. It can be miles and miles deep. But it must let a person enter. And that is what is scary about caves, that one can enter" (Everett 2005, p. 3). The passage points to how our definitions of natural elements are human-centric. The opening to a cave must be large enough to accommodate a human to "qualify" as a cave and, while Hunt suggests that wind and water can enlarge the cave, he later notes that cave-like structures can also be "the result of blasting" (Everett 2005, p. 203), a marker of violent human intervention. Marie-Agnès Gay observes how this opening inaugurates the definitional uncertainty characteristic of most of the characters and their dialogue (10–11), with caves themselves expanding in definition to include metaphorical and emotional aspects: For instance, readers eventually learn that John's now-deceased wife Susie feared the cave, while he and his future wife Morgan use the cave to share their first intimate moment. The "unstable sign" (Gay 2015, p. 11) of the cave is metaphorical of nature itself in *Wounded*, its path cavernous and winding.

John's human-centered perspective supersedes his ability to define the cave on its own terms, revealing his limits in overcoming the hierarchies implied in binaries of human vs. nature. In this sense, *Wounded* reveals John's tendency towards human exceptionalism, which leads him to "understate the degree to which people, animals, artifacts, technologies, and elemental forces share powers and operate in dissonant conjunction with each other" (Bennett 2010, p. 34). John's human-centric definition of the cave also calls to mind the conundrum of considering shared otherness in natural terms. Describing animal alterities, Sune Borkfelt observes, "The quest to know the animal other is... neither easy nor ethically straightforward, given that the ways in which we may go about this are rooted in a firmly human epistemology; our ideas of what it means to 'know' anything, animal or otherwise, will necessarily be human, and any assumption that human forms of knowledge are the only forms would seem to be an anthropocentric notion" (Borkfelt 2016, p. 2). The epistemology and phenomenology represented by John's complications of his definition of the cave incite our need to consider otherness on terms beyond the animal/human binary. His history as a cattle rancher and then a horse trainer speaks to a relationship that is emotional, but certainly also transactional. In one sense, John's close and regular proximity to nature and its creatures puts him in a better place to know them intimately, but his dependence on them for his livelihood tasks him with unknowing them beyond their mere utility. After all, even the most awe-inspiring tasks can lose their luster when relegated to the mundanity of a work routine. At times, John has to make an effort to view interactions with nature as more than items on a to-do list.

In this way, *Wounded* reveals how John's current chosen occupation as a horse trainer both blurs and reinforces the human/nature divide, highlighting the nuances between managing versus working with or even serving nature. His newest horse "client"—a massive palomino named Felony—arrives at the behest of Felony's owner, and John's old friend, Duncan Camp. At first, Duncan and John make fun of Felony's recalcitrant and skittish demeanor; at one point, John jokes, "I say we shoot him" (Everett 2005, p. 23). At the same time, even while entering the pen and jesting that Duncan should get ready to "call nine-one-one" (24), both men are aware of the dangers of working with a wild animal, John likely thinking of his wife Susie, who died while attempting to mount an unwilling and untrained Appaloosa. Later, recalling his mortality while riding Felony, John discovers that the horse responds to his moods; when John is irritated, the horse tenses up, and when he relaxes, the horse calms in response. Felony's responsiveness reflects Catherine Parry's assertion that animals in literature are often simply "characters whose furred or feathered exteriors clothe the concerns, beliefs and interests of humans" (Parry 2017, p. 85). While one might read the connection as one of shared experience, the connection only proceeds in one direction; between horse and rider, Felony proves to be more empathetic.

While the majesty of the surrounding desert inspires awe, Felony's emotional intuitiveness initially aggravates John, revealing his stubbornness in taking the horse's concerns seriously. "All I could do was shake my head. I had to train this horse to tolerate the troubling thoughts of his rider. This was too much" (Everett 2005, p. 58). John's exaspera-

tion may be less with the horse and more with himself: Felony's ability to mirror John's emotions forces John to contend with his personal history, including subjects he has been avoiding. John eventually has a change of heart about this aspect of animal-human inter-action. Toward the end of the novel, a similar moment of cross-species mirroring occurs when his and Gus's abandoned coyote-turned-domesticated dog, Emily, bites David's dad, Howard. Since he'd been lax in training Emily, Gus apologizes, but John shrugs it off: "Emily's fine. . . . She did what she's programmed to do when she's scared. Howard was scared, so she got scared" (Everett 2005, p. 197). John's shift from the earlier scene to the later one shows how much the novel's events have affected John's relationship with animals, particularly as most of the strife he experiences begins and ends with humans.

Nevertheless, John's exasperated reaction to Felony's empathic sensitivities reveals his view of nature as not wholly idyllic but rather tempered by his livelihood and knowledge of nature's brutality. As earlier implied, the term "naturalistic," after all, could simply refer to the level of realism in a work, and diverging from a romanticized version of nature to consider its complexities is more scientifically accurate. Despite this realistic approach, John simplifies his relationship to nature—and specifically the natural American West—when defending his choice to live there. Though discussed primarily in the context of "True Romance," a story in Everett's collection *damned if i do* (2004), Frédéric Dumas's observation of the work as having a "pastoral impulse" also applies to *Wounded* (Dumas 2013, p. 62). John's decision to live with Gus in the Wyoming desert exemplifies what Leo Marx called "the urge in the face of society's increasing power and complexity to retreat in the direction of nature" (qtd. in Dumas 2013, p. 62). John's avoidance issues and his choice to live in the desert in this way seem connected.

An example of John's defensiveness over his choice of retreat occurs when David arrives in Highland with his cantankerous boyfriend Robert to join a rally against the hate crime murder of a gay man, Jerry Tuttle—incidentally very close in time, place, and notoriety to the real-life 1998 killing of Matthew Shepard in a bar in Laramie, Wyoming. Understandably suspicious given recent events, Robert launches a hostile interrogation into John's life, failing to hide his surprise as he learns that John studied art history at Berkeley, and then demanding an explanation of John's choice to live in Highland. Sensing "a tone to his question," John responds, "'Did you notice the landscape when you drove in? [. . .] This is a beautiful place.' I pulled back some. 'I love horses. This is where I grew up. Well, down in Colorado.' I shrugged" (Everett 2005, p. 51). Appealing to the landscape's beauty is the easy answer, one that John eventually complicates after bringing Robert and David back to his home. David finally agrees with John that the scenery is stunning, to which John responds, "'It's work is what it is,' I said. I looked at it. It was especially beautiful under the dusting of snow" (Everett 2005, p. 64). John looks upon his ranch and sees both beauty and labor, the latter evident in his constant reminders to himself to feed or hose down animals, check hooves, maintain bits and harnesses, and so on.

The dual views of nature as both work and beauty seen in the landscape and in John's occupation expose how *Wounded* deals with both the mythical quality of the West (which will be discussed in the next section) and nature's stark realism. Felony can kill his rider with size and brute strength but can also sense and react to John's fear and insecurity. Explaining Felony's demeanor to David, John clarifies, "It's not so much that he thinks he's a person as he doesn't know that people aren't horses. That's a bad thing. Like I said, he's making progress. Or I'm making progress, I should say" (Everett 2005, p. 68). John's correction shows a truism of his occupation: Training horses is less about the horse and more about the person's ability to work with the horse. This is why, in describing the occupation of horse training, Ann Game prefers the mutual term, "entraining" (Game 2001, p. 3), which emphasizes sociality and interdependence across species, a sharp contrast to the term "breaking," a term Parry points out is sometimes used as training's more violent-sounding synonym (Parry 2017, p. 128). Even before working with Felony, John's discussions with Duncan imply John's preference for entraining, as he recommends that

Duncan's daughter should ride Felony in John's presence, allowing John to essentially train the rider alongside the horse.

Like Felony, John is not above seeing a likeness between himself and his animals. In fact, Game characterizes such equivalence as a necessity for good horse trainers, extending the concept of likeness to a sense of oneness between human and animal. Describing horse riding as an art, she relates, "I have come to appreciate just how important a forgetting of our separate human self is if we are to ride well" (Game 2001, pp. 8–9). John's role as both rider and trainer puts a wrinkle in this separation. When readers first meet John, he is using a rasp to smooth an older horse's hoof, which both strengthens and protects the horse's hooves while saving John more work—or even the heartache of prematurely losing a horse—later down the line. The act speaks to John's role as steward and caretaker, but also shows how he benefits from the relationship. His bones cracking, John realizes that he may have asked too much of the horse as well. He may certainly be projecting his mortality onto the horse, but at the same time, his compassion tempers his treatment of both the horse and himself. In this way, much like his complaint about Felony misunderstanding people as horses, John gains advantages from his perceived shared otherness with the creatures around him, each of whom he recognizes as an agent acting of its own volition—whether it is Felony learning companionship, his friend Daniel White Buffalo's mule repeatedly escaping his pen, or his dog Zoe following along on John's various trips into the desert. If John is right that Susie died trying to prove her worthiness and bravery to him by mounting and thus attempting to tame an unruly horse, then Susie's lethal efforts are also misguided, sealing John's belief that trainers must defer to horses. John's work on the horse hoof further reveals how much the natural environment influences his interactions, with John desiring to protect his animals from the harshness of the landscape—the eroding sand, predators, and so on—as much as from the errant rider or cruel handler.

In discussing ways to complicate the human/nature and human/animal binaries, it's important to include Indigenous perspectives, many of which never made these distinctions in the first place. "First of all," Kim TallBear (Sisseton Wahpeton Oyate) points out, "indigenous peoples have never forgotten that nonhumans are agential beings engaged in social relations that profoundly shape human lives" (TallBear 2015, p. 234). Working from Indigenous, queer, and scientific perspectives, TallBear highlights the ways that nonhuman beings have always acted upon us and even extends the breaking of human/nature and human/animal binaries to underline Indigenous cultures' spiritual unifying of the living/non-living binary as well—a perspective implied, for example, in John's heavily metaphorical perception of the cave and his emotional reaction to both animate and inanimate elements of the desert. Frederick Jackson Turner noted in 1893, "The most significant thing about the American frontier is, that it lies at the hither edge of free land. [. . .] the frontier is the outer edge of the wave—the meeting point between savagery and civilization. [. . .] Moving westward, the frontier became more and more American" (Turner 2016, p. 41). Turner's prejudiced binary of savage/civilization reveals his inability to anticipate how study in the humanities has enabled some to come around to the realization of what Vine Deloria, Jr. (Lakota) calls "American Indian metaphysics," which he describes as "the realization that the world, and all its possible experiences, constituted a social reality, a fabric of life in which everything had the possibility of intimate knowing relationships because, ultimately everything was related" (Deloria 2001, p. 2). Though not Indigenous himself (John responds in his usual terse way when, at one point, Gus points out that his grandfather "was a full-blooded Seminole Indian" (Everett 2005, p. 38)), aspects of John's shared otherness with his animals recall the radical relationality that Deloria describes, focused less on domination and more on trust, caretaking, and connectedness.

However, even in the powerful brutality of the desert, human agency does indeed bring about, as Mooallem might put it, changes that hasten nature's disfigurement. In fact, rather than autonomy, sentience, or agency, a more significant dividing line in *Wounded* between humans and animals is wanton malevolence towards others. An example occurs when John is on land that he leased out to the BLM after giving up cattle ranching and

downsizing to fifteen hundred acres ("Not so big" (Everett 2005, p. 52), he tells a disbeliev-ing Robert). In the traditional sense of the word, the land he rides through is as "natural" as could be, relatively untouched and not even used for sheep grazing. John is admiring the red soil when he spots a coyote den that someone had burned out with gasoline while the mother and cubs had been sheltered inside. The mother, dead nearby, has taken the brunt of the burns, attempting to carry the flames out with her to protect her young. Noticing tire tracks nearby that he would eventually attribute to the group of neo-Nazis who move into the town's outskirts, John feels "sick. I was confused, near tears, angry. No one was keeping sheep there, so the lame excuse of protecting stock didn't even make sense" (Everett 2005, p. 45). Together with Gus, John eventually saves one of the pups whose loss of a leg makes her one of the most obvious references to the novel's title.

Grappling with the trauma of discovering and losing most of the terrorized coyotes, John imposes a line between humanity and nature having to do with reckless cruelty. John is aware that sheepherders killed coyotes who threatened their livelihood, and even though he calls it a "lame excuse," he is nevertheless sympathetic. Even when admiring horse and mule bits at the feed store, John admits, "Many were beautiful. All were meant to cause possible discomfort" (Everett 2005, p. 12). In other words, John's apprehension is less about the pain that exists in nature, and more about the cause; if present, discomfort must serve a justifiable purpose. That the terrorizing men—referred to by other characters as neo-Nazis, rednecks, and assholes—kill the coyotes simply for the sake of doing so is what haunts John. Already veering towards misanthropy, he relates, "No doubt because of the coyote, I was hating people more than usual as I drove into town. I drove past the Wal-Mart that I refused to enter, past the McDonald's that I refused to enter and past the church that I refused to enter" (Everett 2005, p. 48). It is no surprise that John and Gus later discover that the neo-Nazis are responsible for both Tuttle's murder mentioned at the beginning of *Wounded*, and the later killing of John's friend David. Nastiness, it seems, does not differentiate between human and animal. Still, John clearly believes animals of the non-human variety have a better moral compass when it comes to wanton killing.

In fact, the presence of these bigoted interlopers seals humankind's role as enacting violence without reason—a trait John describes as all too human, and the one real dis-tinction between human and non-human animals worth contemplating. John first hears the disturbing news of the hate crime from the feed store owner, Myra, who says she "heard [Tuttle] was strung up like an elk" (Everett 2005, p. 12). Myra laments, "You know, people are just animals anymore", to which John responds, "No, they're people. That's the problem." Humans, John implies, are prone to a different kind of savagery than the kind Turner intended when talking about the frontier—one less about bigotry and more about plain brutality. Noticing the store's unsold bicycle chain mule bit that nevertheless "served as a reminder of how cruel people could be" (Everett 2005, p. 13), John defies common associations of humans with sense and reason, instead upholding non-human animals as more reasonable and utilitarian, less likely to be cruel for cruelty's sake. Given this complexity, it's no wonder that John's pastoral impulse always leads him back to the land. Taking Felony for a ride into the desert, John observes, "It was dramatic land, dry, remote, wild. It was why I loved the West. I had no affection necessarily for the history of the people and certainly none for the mythic West, the West that never existed. It was the land for me. And maybe what the land did to some who lived on it" (Everett 2005, p. 45).

## 3. Acting Natural

When considering Everett's *oeuvre* and John's concept of the mythic West, another novel, *Walk Me to the Distance* (Everett 1985), comes to mind. Like *Wounded*, Everett's second novel takes place in Wyoming and features a protagonist, David Larson, who witnesses unhinging and unnatural cruelty perpetrated by humans. *Walk Me to the Distance* further echoes one of *Wounded*'s many murders when David and his counterparts kill a rapist, similar to a murder Gus also commits before the novel's beginning when he killed the man who had raped his wife. The pastor in *Walk Me to the Distance*, while not commenting

directly on the murder, offers a spectrum of immorality that can provide nuance to the levels of discomfort and cruelty that John attributes to people's varied relationships with animals. "There are bad things," the pastor said, "and there are evil things in this world. But a bad thing need not be evil. A bad thing need not be wrong. Many a right action is unpleasant. God judges us as much by our intentions as by our deeds" ([Everett 1985], p. 136). The pastor's view of multiple valences of evil and John's reluctant differentiation between sheepherders and people who kill out of malice topple any dualisms of good versus evil that are so characteristic of classic Westerns. Nevertheless, while some of the events and characters in *Wounded* challenge the common Western binary of good/evil, other aspects of the novel so closely embody the elements of the classic Western as to verge on stereotypical or cliché.

This section thus examines *Wounded*'s connection to the mythical American West and the classic Western genre through an analysis of a second possible application of "naturalistic." Namely, rather than the direct, more realistic references to nature examined in the previous section, I focus here on the word's mimetic and even parodic applications to show how these further complicate what might otherwise have been a simple sense of shared otherness between humans and nature. While the "-ic" suffix does indeed have direct connotations, its meaning can also be a step removed from the original referent. For instance, by describing John as "laconic", I am not only suggesting he is terse in speech; the word further implies the *OED* ([Laconic 2021]) definition of him being "[c]haracteristic of the Laconians; Spartan-like." Similarly, though *Wounded*'s realism does not readily invite supernatural interpretations, we can assume that John would characterize the neo-Nazis—in their murders of man and coyote—as demonic, both because of their malevolence and how their behaviors resemble those of a demon. "Naturalistic" in this sense, then, can point to how characters aspire to, resemble, or even deliberately perform and/or caricature aspects of the storied and not-quite-natural Old West. This resemblance and performance to the Old West calls to mind Jennifer L. Fleissner observation that "nostalgic" forms of naturalism emerged in response to the stresses of modern life, "returning masculine power and adventure to a vitiated modernity by rediscovering the freedoms and struggles associated with a still wide-open, untarnished natural landscape" ([Fleissner 2004], p. 7). While the presence of this Hollywood-infused mythos in *Wounded* might reveal slippage in the novel's otherwise realistic aesthetic, the relationship that the classic Western has to the "actual" West in *Wounded* shows how inextricable the real and surreal have become in this extreme environment. In this sense, the mimetic nature of *Wounded*'s naturalism serves as critique rather than outright nostalgia.

When filming someone, one may direct another to "act natural"—a command to act as they would normally, as though no one were watching. Acting natural has connotations of authenticity or feigned realism. The irony of such a directive is that it often forces one into a self-consciousness that belies acting naturally. The very term, "act natural" suggests performing a behavior that is free of performance, and demands an artlessness and effortlessness that requires conscious effort to pull off properly. Particularly through its protagonist, *Wounded* draws attention to these conscious performances to reveal how the characters interact with these ubiquitous classic Western tropes, even as they recognize some to be farcical or stereotypical. John's reluctance about deep engagement with the world and his constant, often sardonic, meta-analysis of that hesitation as the novel's narrator lends *Wounded* its mimetic quality, with the pacing matching John's droll delivery. For instance, when John's ranch hand, Wallace, gets wrongly accused of Tuttle's murder, John—as Wallace's boss and seemingly closest ally—is called to intercede between him and the police. Anticipating the protests to follow, Sheriff Bucky Edmonds rounds up a higher number of deputies than usual, to which John responds, in his "best cowboy voice," "Are you expecting trouble, sheriff?" ([Everett 2005], p. 32). In fact, John self-consciously deploys his "cowboy voice" on numerous occasions including joking about the weather (p. 68) and flirting with Morgan (p. 128).

More than just an appearance of being a cowboy in the Hollywood sense of the word—with its associations with virility, deliberately performed nonconformity, and unblinking fearlessness in the face of imminent violence, John's performance of cowboy stereotypes further reveals his self-consciousness about human interaction and his awareness of how he is perceived by others—both in ways that evoke the stereotypical Western but also subvert it. For instance, when David's dad Howard and soon-to-be stepmom Pamela show up at John and Gus's home, their observations affirm John's role as a "typical" cowboy even as their pretentiousness reveals them to be, in clichéd cowboy parlance, typical white city folks.

It's worth noting that John's characterization as a standard cowboy is itself a subversion of the stereotype given its whitewashing. Patton and Schedlock note even the term "cowboy" was originally pejorative and reserved only for Black people, "boy" being a diminution of "man" (Patton and Schedlock 2011, p. 507), while an 1878 series of chronophotographical images, entitled "Sallie Gardner at a Gallop" or "The Horse in Motion" feature an unnamed Black man riding a horse in what some might call the first motion picture and what is certainly a "crucial stage in the prehistory of cinema" (Meurer 2014, p. 36). In this way, Black cowboys were not only a prominent part of the Old West, but also present at the birth of Hollywood itself. Or, as Em Haywood (played by Keke Palmer) in Jordan Peele's Nope suggests, "Since the moment pictures could move, we've [i.e., Black people and specifically Black cowboys] had skin in the game." (Nope 2022). Revealing her Hollywood-influenced, citified ways, Pamela looks at the woodfire burning on the stove and comments that it is "like a postcard" (Everett 2005, p. 140). Meanwhile, Howard marvels at the "funny little [gas] station" whose owners are so "trusting" they let you pump gas before you pay, and then hears of their elk stew dinner, and exclaims, "You hear that, Pammy? Elk. We're on the frontier" (Everett 2005, p. 143). The stereotypes of *Wounded*, then, play against each other. Readers know John at this point to be far from stereotypical and know to read Pamela and Howard's banal urban elitism—via John and David's distaste and Gus and Morgan's bemusement—as an intrusion.

Cowboy John and his metropolitan counterparts aren't the only stereotypes whose encounters lend to the attenuation of Western generic conventions. Two different stereotypes run up against each other when Robert, himself consciously performing the role of the aggressively radical gay man, manages to rankle Duncan enough that the latter explains to Robert, in "good old cowboy talk," how the West works. Namely, while Duncan feels this is "a free country" and that Robert is therefore entitled to do as he pleases, "There are people around here who might shoot you for what you are. I don't know why, but I've no doubt that they would. People are bad like that and we've got our share. Maybe we've got more than our share, I don't know" (Everett 2005, pp. 78–79). Duncan thus alludes to a few stereotypes—including the homophobic country bumpkin, the angry bigot who fears what he doesn't understand, and the outspoken cowboy who abides by his own rules—to defend himself against Robert's defensiveness.

Nevertheless, Duncan's nuanced analysis of "good old cowboy" behavior doesn't mean that he too is above parodying classic Western stereotypes for comic effect. Before Robert appears, Duncan is first introduced to David, who John explains is from Chicago. When Duncan's deadpan question of "Where's Chicago?" is met with a blank stare, Duncan tells David, "Just kidding you, son" (Everett 2005, p. 77). A more obvious parody of classic Westerns appears in the feed store, the entrance "always sport[ing] some new, tacky novelty that the shop owner, Myra, hadn't been able to resist. Today it was a pony-sized, stuffed horse with eyes that followed anyone who walked by and said, 'Clippity-clop, cowpoke' in a John Wayne voice" (Everett 2005, p. 12). With his preoccupation with Westerns, it's no surprise that Everett mentions John Wayne in numerous novels, including recent ones such as *So Much Blue* (Everett 2017) and *The Trees* (Everett 2021). In *I Am Not Sidney Poitier* (Everett 2009), Philosophy of Nonsense professor, Percival Everett, asks the title character if he knows Roscoe Lee Browne, adding "he was in *The Cowboys* with John Wayne. I don't much like John Wayne, but Roscoe Lee Browne was great" (Everett 2009, p. 124). The comment

echoes one that the real-life Everett makes in an interview with Robert Birnbaum where he alludes to a film course he teaches on the Western (Everett 2003b), featuring John Wayne whom he "hated" and Wayne's *The Searchers*, which he describes elsewhere as "a movie that at once admits to American racism and practices it" (Klingener 2014). The persona of John Wayne represents the mythic West that John doesn't care for, the swaggering, rugged individualist whose heroism evokes the spirit that supposedly founded the US. While hardened contemporary viewers of *The Searchers* see it as a blatant "white-supremacy parable" (Rose 2020), there remains a contingent even today who romanticize the "Duke" and his movies as the height of American self-confidence and self-invention.

While John is willing to defend the majesty of the landscape that serves as the backdrop for John Wayne's movies, his relationship with the "history of the people" is more complex (Everett 2005, p. 45). In his contentious introduction to Robert, who asks whether John has encountered "problems" with "race," John responds, ""Of course I have, son. This is America. I've run into bigotry here. Of course, the only place anybody ever called me nigger to my face was in Cambridge, Mass.' I let that sink in. 'There are plenty of stupid, narrow-minded people around. They're not hard to find. There are a lot of ignorant people, a lot of good, smart people. Is it different where you come from?'" (Everett 2005, p. 52). When it comes to its human residents, John places the West as just another unexceptional part of a bigoted country, while simultaneously defending it as being not as backward as Robert might assume. Earlier, scanning the news reports of Tuttle's murder, John realizes that "the Eastern papers offer[ed] the implication, if not outright accusation, that the crime was symptomatic of some rural or Western disease of intolerance. I thought, yes, it's called America. I wondered why the reported rash of fifty rapes in Central Park was not considered a similar indicator of regional moral breakdown" (Everett 2005, p. 34). The comments speak to John's problems, not with New England or Massachusetts per se, but with the way that the whitewashed mythos of the West has covered over its more intricate reality, which includes Black, Mexican, and Indigenous cowboys, farmers, and ranchers, and the way that perceptions of prejudice are relegated as a problem assailing mainly rural and backwoods areas and people.

While not self-consciously labeled in the same way as John and Duncan's "cowboy talk," John's acerbic response to Robert of facing bigotry elsewhere carries the same weight of naturalistic performance. The speech is almost miraculous considering John's usually pithy demeanor. As Anthony Stewart notes, John manages to say the right thing at the right time to the right person, "a moment [that] seems only to happen in novels" (Stewart 2020, p. 174). Rather than being prideful, John characteristically expresses regret, feeling "like a bully" and immediately reverting to thoughts of nature to steady himself, admitting to making himself "relax, as when on a nervous horse. I viewed it as good practice" (Everett 2005, p. 52). Much like other moments in *Wounded*, John's pastoral impulse allows him to momentarily suspend the awkward and uncomfortable reality of the moment. Rather than dismissing John's impulse as deflection or avoidance, we might instead align this reaction with what Stewart calls "anti-double consciousness," by which those like John can "insist upon seeing themselves as the multiplicities that they know themselves to be" (Everett 2005, p. 160). At this moment, then, John may be performing for himself, reminding himself of his place in nature and highlighting his complexity as not *just* a Black person who happens to live in the West.

When considering John's speech to Robert, his fusing of New England and the American West in terms of their shared intolerance is also important given that some critics have characterized John's love of the West as unconditional. For instance, Derek C. Maus rightly points out that "Everett gently but unmistakably satirizes Hunt as an unrequited lover of the West; he is willing and able to overlook and to excuse its unsavory aspects, but the favor is never fully returned" (Maus 2019, p. 146). However, when we parse out what Hunt might think of when he thinks of "the West"—the mythic and "real" West, but also the West that is of the natural landscape versus its varied human residents—the love becomes more provisional. As Maus points out (Maus 2019, pp. 145–46), John's differentiation of

Cambridge versus Highland falls flat when the neo-Nazis call both John and later Gus by the n-word (Everett 2005, pp. 200, 202), leading them to act in a way that John notably refers to as "surreal" (p. 199). Gus, in fact, by this point riddled with cancer and worried that David is dead, only allows the neo-Nazi to call him by the slur once before killing him, a recapitulation of Gus's previous killing of the white man who raped his wife. Though Gus was imprisoned—wrongfully, he thinks—for the latter, and may or may not face charges for the former, his act of taking the law into his own hands is a move straight out of a Western.

## 4. The Final Showdown

Our historical inclination to place humans above and separate from animals is steeped in an act of othering that reinforces human superiority. Borkfelt, for instance, notes that "humans have often tended to view those of their characteristics deemed least desirable as betraying our own animality" (Borkfelt 2016, p. 2). John's association of human animals with excessive cruelty and Gus's inclination to deal with this cruelty by enacting justice outside of the law give us insight into the third application of "naturalistic"—related to the American literary movement whose elements include character behaviors that one might call more "animal" than "human", as well as a sense of justice and determinism that might be most appropriately referred to cosmic, related to a lack of free will.

It is worth noting that the narratives commonly associated with American literary naturalism privilege stories of white, ostensibly heterosexual male settlers; Fleissner calls the genre of naturalistic fiction "typically the most hypermasculine in American literary history" (Fleissner 2004, p. 6). Conventional lists of American naturalist writers also sometimes include token mentions of women—namely Edith Wharton and Kate Chopin, while listed Black writers are often solely represented by Richard Wright. Representing naturalist elements of characters falling victim to their social environments, even wealthy white women like Lily Bart of Wharton's *The House of Mirth* and Edna Pontellier of Chopin's *The Awakening* succumb to fateful and devasting defeats despite their willful determination to buck the biological and social limitations placed upon them. Of the former, Larry Rubin notes that Wharton "placed such heavy emphasis upon the molding influence of environment that it seems as if Lily would have had to possess almost superhuman strength to withstand it" (Rubin 1957, p. 183). Donald Pizer nuances this observation by noting that Wharton and fellow naturalist Theodore Dreiser "wish to render the powerful effect of environment and heredity on various specific kinds of temperament and experience" (Pizer 1995, p. 245).

The relationship between African American literature and naturalism is understandably tense given how many white authors at the height of the movement used biology and Social Darwinism to justify the inferiority of Black people and how they "deserved" tragic ends in the Jim Crow era. Nevertheless, scholars like Jericho Williams and John Dudley have expanded the list of Black naturalist authors beyond Wright to include writers such as Claude McKay, Paul Laurence Dunbar, Zora Neale Hurston, James Weldon Johnson, and Ann Petry. Williams notes that for "later writers who experienced, saw, or envisioned some of naturalism's concerns still present and at work. … . the door of literary naturalism never fully closed, especially given that the Jim Crow era and the Great Depression worsened living conditions and experiences for many African Americans after a brief period of progress following the abolition of slavery" (Williams 2021, p. 98). While these authors presented protagonists who were far from being fully at the mercy of their environments, their fates remained intertwined with their surroundings in ways that complicated the hierarchies of humans and nature, as well as determinism as the all-controlling force featured in many naturalist texts.

Particularly striking about Wright's *Native Son* (Wright 1940) being the most cited example of contemporary Black literary naturalism is how Everett's *oeuvre* seemingly rejects the specific form of protest that Wright's work represents, including how critics discussed *Native Son*'s protagonist Bigger Thomas as both a criminal and a victim of his environment. *My Pafology*, the mock-novel at the center of Everett's *erasure* (Everett 2001), is

a work that both parodies *Native Son* and emphasizes that Wright's naturalism remains the unfortunately prevailing marker of so-called Black authenticity. Still, as previously stated, John's positive associations with the West have less to do with its people and more with "what the land did to some who lived on it" (Everett 2005, p. 45), a naturalist characteristic that unites Wright's and Everett's texts. Turner's frontier thesis repeats this naturalist trait if only to debunk it by suggesting how the presumably white man can take over supposedly unoccupied land to create what we now know as the United States, stating, "at the frontier the environment is at first too strong for the man. He must accept the conditions which it furnishes, or perish. Little by little he transforms the wilderness, but the outcome is not the old Europe. . . . [H]ere is a new product that is American" (Turner 2016, p. 41). Turner's characterization of the US as founded on white settler ideals mirrors John's description of an America steeped in bigotry.

Aside from Bigger's eventual fate of incarceration—a question left unanswered in *Wounded*, another major difference between Bigger and John is the latter's socioeconomic status: John's choice to eschew the expectations of his Berkeley education for horse training and his love of fine art reveals access to privilege that interrogates naturalism's emphasis on determinism and lack of choice. In a move that would make Bigger recoil, John even displays a surprising trust in local authorities. He willingly accepts Sheriff Bucky Edmonds' offer to speak to his former ranch hand Wallace when Wallace is accused of murder, calling Bucky "generally agreeable" and "well meaning enough" (Everett 2005, p. 16). When John's Indigenous neighbor, Daniel White Buffalo, is terrorized by the neo-Nazis who kill his livestock and leave racist messages, Daniel and his friend, Elvis Monday, ask John to report the incidents to the sheriff on their behalf. Daniel reasons that "[i]f it comes from just us nobody listens" (Everett 2005, p. 169), but even Gus—who gleefully repeats Daniel's nickname for the sheriff, "Fucky Bucky"—accuses John of being too dewy-eyed, mostly because John doesn't hate Bucky. To that, John asks, "'Why shouldn't I?' I felt on the defensive. I felt that admitting trusting the sheriff was admitting to stupidity or naïveté. Worse, I had the sense that my trusting him suggested a kind of betrayal, but I didn't know of whom. To my question, Gus merely offered a shrug" (Everett 2005, p. 171). John senses that he should be more skeptical, but doesn't feel like Bucky has offered any empirical reason for him to feel that way. Unlike the neo-Nazis, for instance, Bucky has not appeared in John's eyes to enact any unnecessary or excessive violence.

Tuttle's and especially David's murderers—ineptly investigated by Bucky and his counterparts but expeditiously dispatched by Gus and his gun—are what finally force John to see the benefits of active solidarity against oppressive forces such as the police and neo-Nazis, even when John might nonetheless insist on distinct differences between, for instance, racist and queer oppression. Unsure about whether Bucky's incompetence and reluctance to pursue justice is deliberate or unintended, John is forced to reassess his permissive attitude toward authority figures. The neo-Nazis, representing the naturalistic and external evil that invades Highland, also care little to debate the nuances of their hatred of Native or African American people versus their hatred of gay people; after all, they allegedly shot at Elvis Monday's mother, Clara, in addition to killing Tuttle, David, and the coyotes. John's sexuality—fraught even when Morgan brings up the subject—leads to characteristically superficial reflections on queerness, though his thoughts on sexuality (and maybe even Bucky's authority) are unified when you consider them alongside his ideology of letting nature do as nature must. For instance, in his first meeting with David and Robert, he easily concludes that he likes David, but had to try "not to dislike Robert. I wasn't put off by the men's homosexuality, but Robert's display [of his relationship with David] for the benefit of the waitress seemed mean-spirited. I didn't feel bad for thinking that, as I considered I would have been as put off by a heterosexual man or woman similarly marking territory" (Everett 2005, p. 54). John justifies his distaste, then, because he recognizes Robert's excessive territoriality and spitefulness as a cognate to the neo-Nazis' gratuitous actions against human and non-human animals.

The neo-Nazi presence in his hometown thus upholds not just John's theories of humanity's cruelties, but also the aspect of naturalism which involves "[r]evelling in the extraordinary, the excessive, and the grotesque in order to reveal the immutable bestiality of Man in Nature" (Sundquist 1982, p. 13). Neo-Nazis inciting fear and/or anger in Highland's marginalized residents—and, by extension, those who care for and worry over them—further recalls naturalism's questions of whether we really have independence, freedom, and free will. While some authors of the literary movement framed this naturalist question in cosmic terms, Williams and Dudley cite Black authors who bring in man-made sources of determinism such as segregation and Jim Crow laws. Dudley adds, "If naturalist fiction often chronicles the limitations and restrictions imposed on individual freedom, there can be no stronger example of the denial of free will than that imposed by the system of chattel slavery in the United States and the concurrent linkage of a slave's ontological status with legal subservience and inferiority" (Dudley 2012, p. 259). Representing the extremes of racist oppression, the neo-Nazis' terrorism stems from a similar urge to deny the free will of those whom they abhor.

Given the naturalist preoccupation with examining how individual agency fares against oppressive forces and environments, interpreting *Wounded* as a Western-style duel between these two representative groups might make sense. However, the white supremacy that enables the neo-Nazis usually takes a more mundane and thus a more seemingly innocuous form in Highland. Take, for example, Sherriff Bucky's inaction in protecting Highland's more marginalized residents, plus whatever sparks Gus and Elvis Monday's distrust in his authority. The neo-Nazis representing the more blatant and extreme form of this supremacy exist on the outskirts of a town and society that nevertheless are founded on similar principles.

The final showdown in the concluding scenes of *Wounded* further complicates this conflict through its engagement with what is often referred to in Westerns as frontier justice. When David goes missing, John engages in a frantic search for him, rightly convinced that David has been terrorized by the same people who killed Tuttle and the coyotes. Having lost faith in Bucky after the police stop their official search, John follows Elvis's hunch about the neo-Nazis' location, thinking, "Sometimes things were just simple. . .. The people you expected to do the bad thing did the bad thing. I believed the rednecks had done something to David and I was going to find out. Maybe I should have called the sheriff, but I didn't know whom I could trust" (Everett 2005, p. 199). Instead, he grills the men himself on David's whereabouts, finding out almost immediately that they indeed beat up and abandoned David in the desert. Though John raises a rifle to emphasize his questions, Gus is the one who shoots, killing one man in John's presence and leaving the rest for when John takes David to the hospital, where David eventually dies.

Gaining prominence on the eve of American literary naturalism and inspired by the environment of the West, frontier justice represented an alternative to the formal justice system. Under this form of justice, crime was "more a personal affront to the victim than an outrage upon the law of the land. Substantial justice secured in the most direct way was its aim. It had little patience with fine-drawn legal or constitutional distinctions, or even scruples of method. If the thing itself was proper to be done, then the most immediate, rough-and-ready, effective way was the best way" (Turner 1935, p. 20). Given the ways that people like Gus, Daniel White Buffalo, and David find little justice in the legal and official law of the land, frontier justice may seem like an appealing option for the marginalized people of Highland. However, Richard Slotkin points out how members of the political and social elite have also used this concept—like the myth of the frontier itself—to showcase the United States as an egalitarian "land of unlimited opportunity" and to ultimately "blind us to the consequences of the industrial and urban revolutions and to the need for social reform and a new concept of individual and communal welfare" (Slotkin 2000, p. 5). Or, as John might put it, those with the privilege to enact frontier justice do so to be more like people, and less like animals.

## 5. Conclusions

Discussing works by naturalist authors like Jack London, Jeanne Campbell Reesman laments that naturalism's focus on objectivity and impartiality had great potential to address marginalized experiences, but that its practitioners fell victim to pseudo-scientific beliefs in eugenics and social Darwinism instead. "Naturalism," she notes, "was uniquely suited to address the racial debates of the era especially in terms of beliefs in inborn criminality" as well as issues like miscegenation and immigration. "Unfortunately, instead of treating race or criminality objectively, most naturalists went along with popular beliefs in hereditary determinism that fueled both racism and racialism" (Reesman 2012, p. 275).

Though John does not meet the fate of Bigger or any number of Black protagonists in naturalist literature, it's difficult to pin down how John is transformed by the monumental events of the novel's end. Beyond the fact of Gus's cancer, his uncle's fate is uncertain. Meanwhile, despite Morgan's presence, John will likely continue to give in to his pastoral impulse, seeking detachment from the world of human savagery through frequent escapes into nature. At the start of the novel, he is unable to imagine anyone killing Tuttle and ends with disbelief that "Gus had shot that man" and "that I had put myself in a place where I could have shot him" (Everett 2005, pp. 67, 203). On the one hand, we can view John's sustained incredulity as naive, but on the other hand, John may remain stubbornly insistent that humans can learn from the natural environment that surrounds them and that they do not necessarily have to give in to their most human and cruel inclinations.

*Wounded*'s multiple perspectives on the naturalistic offers fruitful possibilities. For one, naturalism finds new life when engaged with contemporary literary approaches such as animal studies, ecofeminism, object-oriented ontology, and new materialism, and vice versa. As Trumpeter notes, such theoretical combinations have the power to "extend ethical consideration and political voice" to entities previously and largely overlooked (Trumpeter 2015, p. 226). Additionally, John's sometimes distanced, sometimes hyperreal perspectives on the West offer ways to reconsider the power dynamics of this "wild" land in both history and the present, especially as it intersects with considerations of Black masculinity. Given the ways that nature has been wholly affected by human interventions, texts such as *Wounded* provide opportunities for readers to consider human stewardship in natural contexts.

*Wounded*'s ending leaves open the question of whether fate and determinism eventually work in John's favor. Instead, readers are left with Elvis Monday's words to John as he returns an exhausted Gus to the ranch: "This is the frontier, cowboy.... Everyplace is the frontier" (Everett 2005, p. 207). Putting these words in the voice of one of the novel's two prominent Indigenous characters reminds us that native people are still fighting to maintain sovereignty and reclaim severed connections to the land, in the West and beyond. Monday's statement further functions, as Maus points out, as "a commentary on the ubiquity of violence claiming to be justice" (Maus 2019, p. 148), a declaration that validates neither frontier justice nor its more officially recognized counterpart. While the ending may frustrate those who would hope for more closure in a modern-day work of naturalism, or even more fortitude in a modern-day protagonist like John, the open ending is itself a statement, a way of refusing to reveal which form of individualist expression wins out, or whether heredity, external forces, and/or the environment ultimately controls the characters and their actions. As Link states when talking about naturalism, "Determinism is a question not a statement. Evolution is a religion that offers salvation and damnation. Human nature is both knowable and clouded in mystery" (Link 2012, pp. 88–89). Whether the frontier will bring forth its intended ideals of egalitarianism and equality remains to be seen.

**Funding:** This research received no external funding.

**Institutional Review Board Statement:** Not applicable.

**Informed Consent Statement:** Not applicable.

**Data Availability Statement:** Not applicable.

**Conflicts of Interest:** The authors declare no conflict of interest.

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
