# Peer review of "Naturalistic Elements in Percival Everett’s Wounded"

_humanities, doi:10.3390/h12050095_

Round 1
Reviewer 1 Report
This is a very good essay which I really enjoyed reading. It is original, engaging, and stays close to the literary text. Its weakness is a that it is at times somewhat intuitively structured, relying on what amounts to elegant plot summary and plentiful examples, rather than more robust claims and support. It is not a big problem, but some more attention to organization and previous research would be welcome. Below, I point to a few such weaknesses that the author may want to address.
1.The author places great weight on the word "naturalistic," (used by Everett in an epigraph which unfortunately does not appear in my copy of the article), and comes back to three possible uses or interpretations of it. However, given the word “nature’s” problematic semantics, the three definitions are somewhat inadequate and the first one, especially, comes across as a little naïve. I would recommend the author to briefly signal awareness of this problem, and then find a source to cite for usage 1 and usage 2.
2. Taken at face value, though, these three uses of the word form a good structuring device, and each section is interesting and relevant. When we get to the conclusion, however, it focuses almost entirely on the author’s third meaning, referring to the literary school of naturalism, and neglects to conclude, at least explicitly, what the other two meanings contributed to the final analysis. This is a pity, and, I think, a missed opportunity to tie these three meanings and their applications in the novel together. I would like to see a couple of sentences in the conclusion that remind us more explicitly also of the findings in sections 1 and 2.
3. Both sections 1 and 2 seem a little under researched, though in different ways. In section 1 the theoretical basis is there, for an informed and interesting analysis of how the novel represents human/animal relationships. However, there is very little, discussion with current similar research (e.g. literary animal studies) on other works in the same or similar genres, which would be welcome. A few such sources could perhaps be included.
3. In section 2, there is no theoretical foundation for the use of the term naturalistic in its mimetic or parodic sense, apart from an OED reference which does not include the semantic and political problems with the words nature, natural, and naturalistic. There are some examples of current research, but most of the sources cited provide not a definition, or a way to pinpoint the usage of the term in this essay, but are various loosely related examples which are similar in style to the author's loosely organized ones, that form the argument rather than support it. A few claims, directing the reader’s attention to what the examples show, would be a great improvement.
4. The third section is better theorized and researched, although the strong connection between literary naturalism and animals (both the inner beast and actual animals) could have been put to better use and could have been cited to support claims such as the one on line 496 about “character behaviors that one might call more ‘animal’ than human’”. This would also have formed a much stronger connection between section 1 and section 3. A few citations would suffice here.
Author Response
Hello and thank you for your feedback! Here is a summary of my revisions:
---Added epigraph
---Signaled awareness of superficiality of first definition at face value per request.
---Included additional sources (including Eric Carl Link, Sarah Balkin, and Jennifer Fleissner) defining naturalism to support both usage #1 and usage #2 in the introduction and body of the article.
---Cited additional sources (including Kevin Trumpeter and Jane Bennett) draw attention to connections requested between this article, New Materialism, and OOO.
---Cited additional sources (including Catherine Parry and Sune Borkfelt) that draw attention to connections between this article and literary animal studies.
---Cited additional sources (including Jordan Peele and Ulrich Meurer) that provide additional historical and contemporary contexts of representations in the West.
---Included more signposting of arguments alongside examples.
---Added additional summative and reflective comments to conclusion
---Streamlined and added additional texts and corrections
Thanks again for your generous and helpful comments.
Reviewer 2 Report
comments are attached

Author Response
Hello and thank you for your feedback! Here is a summary of my revisions:
---Added epigraph
---Added and highlighted epigraph.
---Included additional sources (including Eric Carl Link, Sarah Balkin, and Jennifer Fleissner) defining naturalism to support both usage #1 and usage #2 in the introduction and body of the article.
---Cited additional sources (including Kevin Trumpeter and Jane Bennett) that draw attention to connections requested between this article, New Materialism, and OOO.
---Cited additional sources (including Catherine Parry and Sune Borkfelt) that draw attention to connections between this article and literary animal studies.
---Cited additional sources (including Jordan Peele and Ulrich Meurer) that provide additional historical and contemporary contexts of representations in the West.
---Added clarification to Jericho Williams' assertion (not mine!) that "the door of literary naturalism never fully closed" per request
---Included more signposting of arguments alongside examples.
---Added additional summative and reflective comments to conclusion
---Streamlined and added additional texts and corrections
Thanks again for your generous and helpful comments.
Reviewer 3 Report
I am happy with the paper as it is and have no substantive suggestions for the author/s. I note that the epigraph mentioned in the first sentence is not present in the text as I received it - the author should check that this will be included, or amend the sentence.
Author Response
Hello and thank you for your feedback! I have added in the epigraph. Thank you for catching that. Take care, and wish you all the best.
Reviewer 4 Report
The article is well constructed and developed. The argument is convincing and supported by both close reading of the novel and relevant secondary texts. The only small issue I had was with the introduction, which seemed a bit confusing. The author begins by referencing an "epigraph" and an "interview" that are not given or referenced anywhere. I think a clearer opening paragraph would help to situate the reader from the start.
Author Response
Hello and thank you for your feedback! I have added in the epigraph. Thank you for catching that. I have also added clarifications to the introduction section per your feedback. Take care, and wish you all the best.